# Impact of maternity waiting homes on facility delivery among remote households in Zambia: protocol for a quasiexperimental, mixed-methods study

Nancy A Scott,[1] Jeanette L Kaiser,[1] Taryn Vian,[1] Rachael Bonawitz,[1,2] Rachel M Fong,[1] Thandiwe Ngoma,[3] Godfrey Biemba,[4] Carol J Boyd,[5] Jody R Lori,[6] Davidson H Hamer,[7] Peter C Rockers[1]

For numbered affiliations see end of article.

**Correspondence to**
Dr Nancy A Scott;
nscott@bu.edu

## ABSTRACT

**Introduction** Maternity waiting homes (MWHs) aim to improve access to facility delivery in rural areas. However, there is limited rigorous evidence of their effectiveness. Using formative research, we developed an MWH intervention model with three components: infrastructure, management and linkage to services. This protocol describes a study to measure the impact of the MWH model on facility delivery among women living farthest (≥10 km) from their designated health facility in rural Zambia. This study will generate key new evidence to inform decision-making for MWH policy in Zambia and globally.

**Methods and analysis** We are conducting a mixed-methods quasiexperimental impact evaluation of the MWH model using a controlled before-and-after design in 40 health facility clusters. Clusters were assigned to the intervention or control group using two methods: 20 clusters were randomly assigned using a matched-pair design; the other 20 were assigned without randomisation due to local political constraints. Overall, 20 study clusters receive the MWH model intervention while 20 control clusters continue to implement the 'standard of care' for waiting mothers. We recruit a repeated cross section of 2400 randomly sampled recently delivered women at baseline (2016) and endline (2018); all participants are administered a household survey and a 10% subsample also participates in an in-depth interview. We will calculate descriptive statistics and adjusted ORs; qualitative data will be analysed using content analysis. The primary outcome is the probability of delivery at a health facility; secondary outcomes include utilisation of MWHs and maternal and neonatal health outcomes.

**Ethics and dissemination** Ethical approvals were obtained from the Boston University Institutional Review Board (IRB), University of Michigan IRB (deidentified data only) and the ERES Converge IRB in Zambia. Written informed consent is obtained prior to data collection. Results will be disseminated to key stakeholders in Zambia, then through open-access journals, websites and international conferences.

**Trial registration number** NCT02620436; Pre-results.

### Strengths and limitations of this study

► To the best of our knowledge, this is the first large-scale impact evaluation of maternity waiting homes (MWH), employing a rigorous controlled before-and-after, quasiexperimental design and using mixed methods.

► For generalisability, a representative sample of recently delivered women living most remotely is selected using a multistage, random sampling strategy for both the quantitative household surveys and the qualitative in-depth interviews.

► Half of study clusters could not be randomly assigned to either the intervention or control group due to political constraints, resulting in quasiexperimental study design.

► Because remote women stand to benefit the most from the MWH model, eligibility is limited to those living at least 10 km from the health facilities; findings will therefore not be able to assess impact of the intervention on women living nearer to facilities.

► In companion protocols, implementation fidelity of the core elements of the MWH model is assessed by each partner using harmonised tools.

## INTRODUCTION

The Sustainable Development Goals (SDGs) include a target of reducing the global maternal mortality ratio (MMR) to less than 70 deaths per 100 000 live births by 2030.[1] Zambia's MMR is currently 398 deaths per 100 000 live births, well above the SDG target.[2 3] Skilled care at every birth, one of the two SDG indicators for MMR, is recommended. What remains unanswered is how to best facilitate access to intrapartum and postpartum care, particularly in rural and remote areas where distance and poor transportation severely restrict access to care. The Government of the Republic of Zambia (GRZ) is committed to improving maternal health and

**BMJ** 1

encourages facility-based delivery for all women,[4 5] though accessing facilities for birth is challenging for women living in remote areas.[6–9]

Maternity waiting homes (MWH) are lodgings located near health facilities where mothers who are close to term can await delivery. These homes are meant to provide pregnant women with the option of planning ahead and travelling to health facilities well before labour begins. MWHs may be a promising strategy to improve access to facilities for delivery, but the evidence is mixed. While some evidence suggests they are associated with higher rates of facility delivery and improved maternal health outcomes,[10–20] a Cochrane review found that there are no randomised or quasirandomised trials assessing the effectiveness of MWHs in low-resource settings.[21] Additionally, it is unclear if MWHs can increase access to facility delivery among women living most remotely.[19 22] Rigorous evidence on the impact of MWHs on facility deliveries is needed.

This protocol describes a study being conducted by the Maternity Homes Alliance (MHA), a partnership between the GRZ, Boston University and Right to Care Zambia, formerly the Zambian Center for Applied Health Research and Development (BU/RTC), Africare and the University of Michigan (Africare/UM), and funded by Merck Sharp and Dohme for Mothers, the Bill & Melinda Gates Foundation, and The ELMA Foundation. The MHA hypothesises that MWHs can remove the distance barrier and increase access to facility-based delivery. In this study, we test the impact of MWHs on facility delivery among women living at least 10 km from health facilities in rural Zambia.

MWHs have the potential to improve access to facility delivery, particularly for women in rural areas living far from health facilities. Despite their widespread use in low/middle-income countries, there is currently little evidence of MWH effectiveness. Using community input, we developed an MWH model and are evaluating it for impact. To the best of our knowledge, this is the first large-scale impact evaluation of MWHs. Findings will generate evidence surrounding the effectiveness of MWHs on improving facility deliveries for remote populations in Zambia and other countries with similar rural and highly dispersed populations.

## Intervention

While the GRZ supports the use of MWHs as a strategic method to increase access to skilled birth attendance[5 23] and MWHs have existed in Zambia for decades, there is no specific policy or plan for the scale-up of MWHs and their general quality remains low.[13 24–26] MWHs have been largely constructed through community initiatives or international donors, with limited support for their long-term maintenance.[24–26] Formative evaluations conducted previously by members of the study team in the current study setting showed that MWHs could be an acceptable and feasible option to improve access to facilities for delivery.[24–26] Informed by these findings, the core MWH model was designed to be responsive to community

expectations, community-defined standards of acceptability and community perceptions of quality including safety, comfort, management and services offered (figure 1). In direct response to the formative data, the model includes the following:

► Infrastructure, supplies and equipment: The core MWH model has concrete walls and floors, roofs that do not leak, latrines, a private bathing space, water within a reasonable distance, a covered cooking space and storage space. For safety, the core MWH model has lockable doors, windows, cupboards and lighting. Amenities include beds, mattresses, bedding, mosquito nets and cooking utensils.

► Policies, management and finances: The core MWH model is community owned and operated, as requested by the Ministry of Health. The policies, management and financial structures are adaptable to site-specific needs and preferences, though all have a formalised governance and management structure with community, government and health facility representation. Each also has a management unit responsible for daily operations including registering and orienting women, record keeping and maintenance.

► Linkages and services: Each core MWH model is situated close to the health facility to ensure timely access to clinical care when a woman's labour begins. A health facility staff provides daily check-ins with waiting women, though clinical care visits continue to be conducted at the health facility, not in the MWH. Women staying at the MWH have the opportunity to participate in maternal and child education courses offered by the health facility staff or community health workers.

The core MWH model is promoted in the community through several mechanisms. First, health facility staff promote the MWH at all ANC visits. Over 95% of women attend at least the first ANC visit, so most women are exposed at the health facility.[2] Second, Safe Motherhood Action Group members promote the use of MWHs during their routine outreach activities. Lastly, the traditional leadership (chiefs and headmen) actively promotes the use of MWHs at their community meetings. The core MWH model targets all pregnant women within 1–2 weeks of their estimated delivery date resident within the catchment area, prioritising those women living farthest away (ie, >10 km from the health facility). The 20 MWHs opened in phases between mid-2016 and mid-2017.

## METHODS
### Evaluation questions
The primary research question is:

1. What is the impact of the MWH model on the probability of facility delivery among mothers living more than 10 km from the health facility?

Secondary questions include:

1. Do awareness and perceptions of health facility delivery and health facility delivery intention among

## Core Maternity Waiting Home Model

### INFRASTRUCTURE, EQUIPMENT & SUPPLIES

- Lighting (lanterns)
- Lockable doors, windows
- Cooking area and supplies
- Bathing and laundry areas
- Latrines
- Beds, bedding and bed nets
- Staff room (for storage, office, etc.)
- Space for postnatal women/newborns to stay
- Functional equivalence: concrete floors, no leaky roofs and availability of water

### POLICIES, MANAGEMENT & FINANCES

- Formalized management structure with government and facility representation
- Clear definition of ownership (land, material assets, income generated)
- Revenue and asset management
- Standard operating procedures (SOPs) for clear roles and responsibilities
- Mechanism for community/women's feedback
- Intake, registration and monitoring procedures
- Eligibility: prioritize women living > 10km from health facility, available for postnatal stays

### LINKAGES & SERVICES

- Adjacent to BEmONC, within 2 hours of CEmONC facility
- Daily check-ins by facility staff
- ANC and PNC visits conducted at health facility
- Emergency transport system identified
- Family planning/post-partum family planning education
- Breastfeeding and infant and young child feeding education
- Education on newborn danger signs and well-baby care
- Education on antenatal and postpartum period
- Entertainment and recreational activities

**Maternity waiting homes will NOT provide clinical care:
ANC and PNC Visits will still be conducted at the health facility**

**Figure 1** Core maternity waiting home model developed by the Maternity Home Alliance for intervention sites (n=20). ANC, antenatal care; BEmONC, basic emergency obstetric and neonatal complications; CEmONC, comprehensive emergency obstetric and neonatal care; PNC, postnatal care.

pregnant women living in communities located more than 10 km from the health facility change over time in MWH model sites?

2. How do awareness and perceptions of MWHs by communities located more than 10 km from the health facility change over the period of this study?

3. What financial impact does the use of the MWH model have on the families of women who use it?

4. How does the perception of quality and acceptability differ between MWH model sites and comparison sites?

5. What is the impact of the MWH model on maternal and neonatal health outcomes among those living more than 10 km from the facility?

### Study setting

This study began in March 2016 and will be completed in December 2018. The intervention and comparison sites are located in the primarily rural Zambian districts of Choma, Kalomo and Pemba districts of Southern Province; Nyimba and Lundazi districts of Eastern Province; and Mansa and Chembe districts of Luapula Province (figure 2).

Choma district has a population of 247 860 and a population density of 34/km$^2$, with 68.7% of its population being rural. Kalomo district has a population of 258 570 and a population density of 17.2/km$^2$, with 91.8% of its population being rural.[27] Nyimba district has a population of 85 025 and a population density of 8.1/km$^2$, with 91% of its population being rural. Lundazi district has a population of 323 870 and a population density of 23/km$^2$, with 95.1% of its population being rural.[28] Mansa district has a population of 228 392 and a population density of 23.1/km$^2$, with 61.9% of its population being rural.[29]

### Study design

This study employs a quasiexperimental controlled before-and-after (CBA) design with a total of 40 study clusters, 20 intervention and 20 control clusters. Clusters consist of health facilities and their catchment households. Intervention clusters are receiving the core MWH model, inclusive of newly constructed homes with the elements from the three domains: (1) infrastructure, equipment and supplies; (2) policies, management and finances; and (3) linkages and services detailed in the intervention section of the protocol. Control clusters are implementing the 'standard of care' for waiting mothers in Zambia. Because no national policy exists, the standard of care is facility driven and varies widely. Some

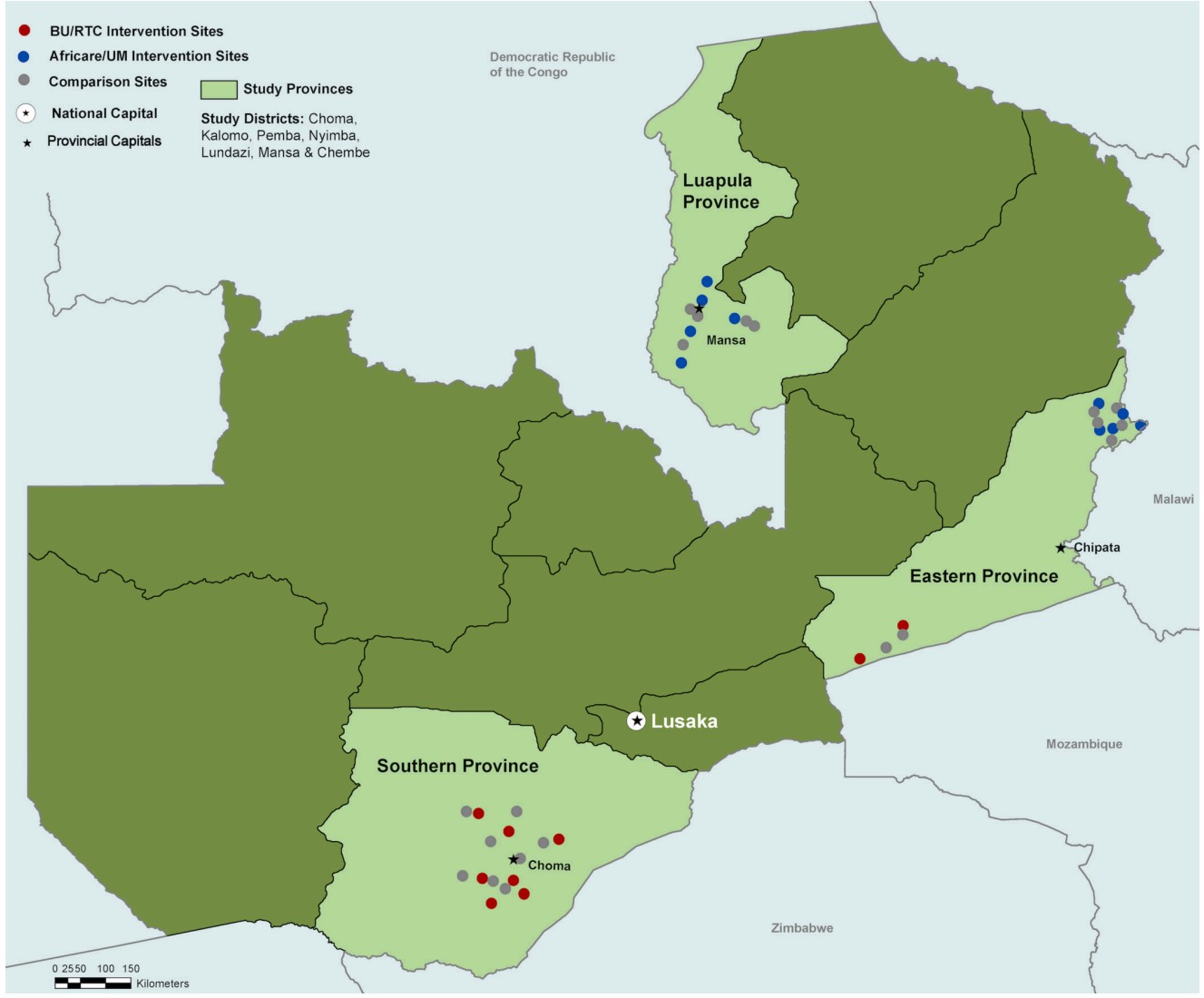

**Figure 2** Map of the Maternity Home Alliance intervention and control study sites by partner. BU/RTC, Boston University and Right to Care Zambia; UM, University of Michigan.

standard-of-care facilities have no designated space for a mother to wait; others have no MWH but provide a designated space for waiting mothers within the clinic; and a small number have an existing MWH-like structure but with highly variable quality.[13]

### Eligibility criteria of study clusters

Because the intervention aims to generate demand for health facility delivery, it is critical that facilities are capable of managing basic emergency obstetric and neonatal complications (BEmONC). Because of inconsistencies in available secondary data sources across the different districts, we established supplemental criteria that could be drawn from the available sources.[30 31] Clusters were eligible for inclusion in the study if the health facility was located ≤2 hours driving time to a comprehensive emergency obstetric and neonatal care (CEmONC) capable referral facility, performed a minimum of 150 deliveries per year and met at least one of two sets of conditions below:

Eligibility condition set 1:
i. Facility is able to provide at least five of seven BEmONC signal functions based on 2015 data.

Eligibility condition set 2:
i. Facility has at least one skilled birth attendant on staff.
ii. Facility routinely provides active management of third stage of labour.
iii. Facility has had no stock-outs of oxytocin in the last 12 months.
iv. Facility has had no stock-outs of magnesium sulfate in the last 12 months.

### Selection and assignment of study clusters to study arm

There is a total of 40 clusters (20 intervention, 20 comparison) in this study (table 1). Each implementing partner used different methods to select and assign clusters to study arms. BU/RTC supported areas had a total of 36 eligible health facilities that were located ≤2 hours driving time to a referral facility and performed a minimum of

**Table 1** Quasiexperimental study design to evaluate the impact of MWHs in rural Zambia

| Randomised subsample (n=20 clusters) | Non-randomised subsample (n=20 clusters) | Non-randomised full sample (n=40 clusters) |
|---|---|---|
| R O1 X O2 | NR O1 X O2 | NR O1 X O2 |
| R O1 _ O2 | NR O1 _ O2 | NR O1 _ O2 |

MWH, maternity waiting home; NR, not randomised; O, observations at baseline (O1, in 2016) and endline (O2, in 2018) at intervention (X) and comparison (_) sites; R, cluster randomised; X, minimum core maternity home (see above).

150 deliveries per year. Of those, 22 (61%) met one of the two eligibility conditions. This partner selected the 20 farthest away from referral, created 10 pairs matched on annual delivery volume and distance, then randomised matched pairs to intervention or control, using the RAND function in Microsoft Excel. All eligible sites were included regardless of the presence of an existing infrastructure or space that functioned as an MWH. Control sites with existing infrastructure or space are considered standard of care. Though sites with existing MWH infrastructure were generally not structurally sound.

Africare/UM had a total of 29 eligible health facilities that were located ≤2 hours driving time to a referral facility and performed a minimum of 150 deliveries per year. Of those, 22 (76%) met one of the two sets of eligibility conditions. Africare/UM was unable to randomly allocate sites to a study arm due to local political constraints, as the Ministry of Health feared community fatigue due to the large number of organisations implementing projects and conducting research. They instead worked with the Ministry of Health to identify 10 intervention sites using the same eligibility criteria. They then selected comparison sites, matched to intervention sites on annual delivery volume and distance to a referral hospital. Sites with an existing infrastructure that functioned as an MWH were not considered as an option for comparison sites. After selecting sites, both partners then constructed the core MWH model at each of the 20 intervention sites.

## Data sources

Population data are being collected from two main sources: household surveys (HHS) and in-depth interviews (IDI). Baseline data collection occurred in early 2016 prior to the implementation of the MWH model in intervention clusters; endline data collection will occur in late 2018, after an 18-month intervention period. The HHS is administered to a sample of 2400 recently delivered women (eligibility criteria described below) residing in intervention and control clusters. In the case of maternal death, the household head or senior woman was interviewed as a proxy participant.

The HHS captures information on the domains and data fields seen in table 2. The HHS was pretested among a sample of 50 participants representing all the major local languages. At baseline, only small adjustments were made in response to the pretest, primarily changing formal translations into the vernacular.

IDIs are conducted among a subsample of 240 HHS participants in order to gain a deeper understanding of community awareness, perceptions and experiences. Because the seven districts are spread out and culturally different, we wanted to ensure we reached saturation or predictability in each district to better explore context with the qualitative data.[32] Consequently, we planned to conduct a large number of IDIs to make sure there was sufficient coverage of different populations to provide insight into the quantitative survey findings. IDI content builds on themes captured in the HHS and includes perceptions of labour and delivery practices, barriers to accessing care, knowledge and awareness of MWHs, sources of knowledge of MWH, perceptions of the quality of maternity homes (safety, comfort, management and services), perceptions of MWH ownership, perceptions of health facility and expenses incurred for last delivery.

The population-based approach captures the experiences of those who used the facility in their catchment, other facilities and those who did not access a facility for delivery, allowing us to more accurately estimate the impact of the MWH model intervention among women living farthest from the health facility in an intention-to-treat analysis.

## Sampling strategy and sample size

To estimate the impact of the MWH model based on an intention-to-treat analysis, we aim to select a representative sample of women in our sample frame who delivered a baby in the past 12 months, irrespective of her place of delivery or her use of an MWH. With this strategy, we will also be able to explore the relationship between use of the MWH and location of delivery. As such, we are recruiting a repeated cross section of 2400 households at each round of the survey (approximately 60 households per cluster): 1200 from both intervention and control sites at both baseline (completed in 2016) and endline (planned for 2018), for a total study sample of 4800 households (table 3).

After accounting for the clustered sampling design (intracluster correlation coefficient estimated at 0.04 based on previous work[33–35]), and assuming an alpha of 0.05, this sample will provide us with 80% power to detect a minimum 10 percentage point difference in the anticipated impact of the MWH intervention on the primary outcome of facility delivery, a programmatically meaningful difference. We recruited a sample of 240 women for the IDIs (randomly selecting 10% of the household sample) at baseline, and will recruit another 240 at endline.

## Participant recruitment

For the purposes of this evaluation, a household is defined as a group of people who regularly cook together. Inclusion criteria for the HHS are:

**Table 2** Summary table of data fields collected from the household survey

| | |
|---|---|
| Household panel | ► Geo-coordinates of household/distance from nearest health facility<br>► Age and sex of household members<br>► Education level of household members<br>► Recent pregnancy/delivery of household members |
| Individual demographics and household characteristics | ► Number of pregnancies<br>► Outcome of pregnancies<br>► Number of living/deceased children<br>► Characteristics of living quarters (eg, roof type, floor type, cooking fuel type)<br>► Access to and quality of water<br>► Household wealth indicators and assets |
| Last pregnancy | ► Antenatal care services used<br>► HIV testing<br>   – Status at last pregnancy<br>   – PMTCT services<br>► Perceived satisfaction with antenatal care |
| Last delivery | ► Location of last delivery<br>   – Decision making around location for delivery<br>   – Mode of transportation<br>► Referral and bypassing<br>► Receipt of CEmONC services (C-section, blood transfusion, intravenous antibiotics)<br>► Perceived quality/satisfaction with delivery services<br>► Maternal and neonatal vital status |
| Use of MWH | ► Knowledge of MWH<br>► Source of knowledge of MWH<br>► Nearest MWH to home<br>► Use of MWH before/after last delivery<br>   – Cost of using MWH<br>   – Perceived quality of MWH (safety, comfort, management and services)<br>   – Satisfaction with MWH<br>► Use of MWH for other pregnancies or other maternal health visits<br>► Intended future use of MWH |
| Cost of delivery and delivery planning | ► Planned or intended location for delivery<br>   – Adherence to planned or intended location for delivery<br>► Barriers to birth plan adherence<br>► Savings for last delivery<br>► Cost of last delivery (broken down by expense) |
| Postnatal care (PNC) and infant health | ► Time to first maternal and newborn postnatal visit after delivery<br>► Perceived quality of postnatal services received<br>► Breast feeding practices<br>► Supplementary feeding practices<br>► Newborn vaccination status<br>► PMTCT/ART for newborn<br>► Interactions between the parent and the child<br>► Maternal depression assessment<br>► Health-seeking behaviour for child's last illness |
| Healthcare knowledge and beliefs | ► Use of contraceptives for family planning<br>► Primary barriers to accessing healthcare<br>   – Primary barriers to accessing skilled delivery services |

ART, antiretroviral therapy; CEmONC, comprehensive emergency obstetric and neonatal care; C-section, caesarean section; MWH, maternity waiting home; PMTCT, prevention of mother-to-child transmission of HIV.

► Household with someone who has delivered a baby within the past 12 months, irrespective of maternal or infant vital status.
► Participant must be age 15 or older. If aged 15–17, a legal guardian must be available for consent.
► Proxy participant (if woman deceased) must be over the age of 18.

► Resident of the village identified for sampling (≥10 km from the facility).

To select a sample representative of women living at least 10 km from our health facility, we employ multistage random sampling procedures (figure 3). We begin the first stage of sampling by visiting every village within the catchment area of each study site, informing the local

**Table 3** Total sample size for evaluation

| Evaluation activity | Intervention sites | Comparison sites | Households per site | X2 observations (baseline and endline) | Total |
|---|---|---|---|---|---|
| Household survey | 20 | 20 | 60 | 2 | 4800 |
| In-depth interview* | 20 | 20 | 6 | 2 | 480 |
| Total participants for all evaluation activities | | | | | 4800 |

*In-depth interviews (IDI) are a subset of the total household survey population selected for more in-depth information and are therefore not factored in as additional human subject participants in the total sample size for this study.

village leader of the purpose of the study and taking the global positioning system (GPS) coordinates from the approximate geographical 'center' of the village. We input these GPS coordinates into ArcGIS Online (Esri, Redlands, CA) and use the line creation tool to draw the most direct route along the roads and paths visible on the World Imagery basemap between each village centre and their associated health facility. We then use this network of roads to calculate the distance of each village to the health facility and develop a sampling frame of all villages within each catchment area located more than 10 km from the health facility (rounding up from 9.5 km). We then randomly select a sample of 10 villages from each catchment area with probability proportional to population size. We list every eligible village within a catchment area in Microsoft Excel along with the total population of the village. We assign a series of numbers to each village, corresponding to the population size (ie, if village 1 had 30 people, 1–30; village 2 had 20 inhabitants, 31–50), and use the random number generator function to select the villages in each catchment area.

Second, we work with community volunteers and village leaders to list all households within the selected villages that have a woman who had a delivery in the last year. We randomly order them by rolling a dice twice, first for a random start and then for a random skip until all households are ordered. We visit each household in that order and confirm their eligibility for study participation. We continue down the list until six eligible households in each village are identified. We select additional villages and additional households if necessary to reach our sample of 2400 households per round. This process assumes that the health facility staff are able to accurately and completely identify all villages within their catchment area.

The study team and community volunteers introduce the study to potential participants and request permission from the household head or most senior woman in the household to screen for eligibility. If household eligibility is confirmed, the study team proceeds with the informed voluntary consent process with the household head or senior woman. Once informed consent is obtained and

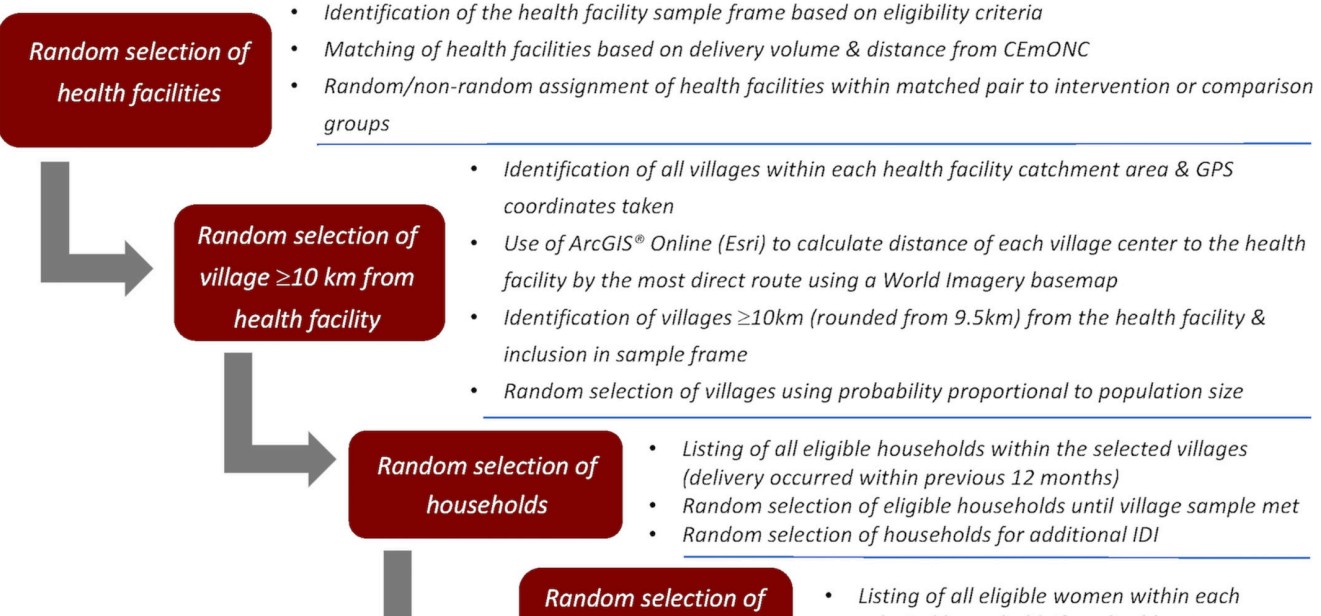

**Figure 3** Multistage random sampling strategy for baseline and endline. CEmONC, comprehensive emergency obstetric and neonatal care; GPS, global positioning system; HHS, household survey; IDI, in-depth interview.

documented from the household head or senior woman, the enumerator records the geolocation of the household and commences the interview or schedules a later appointment. The household head or senior woman responds to the first part of the survey for approximately 15 min, enumerating all of the people in the household in a table that captures demographics as well as recent deliveries and delivery outcomes.

On completion of the household demographics and enumeration, an eligible woman is selected to respond to the remainder of the survey. If more than one woman in the household had delivered a baby in the past 12 months, the electronic data capture system is programmed to randomly select one eligible woman to respond to the remainder of the survey. The selected woman is then consented separately, enrols in the study and completes the HHS in a private space where she feels comfortable. Completion of the HHS takes approximately 45 min.

Of the woman participants, 10% are randomly selected to participate in a 30 min IDI immediately following the survey. IDI participants can take a short break after the HHS, or reschedule if more convenient. The household-level sampling procedures described here have been conducted at baseline (2016) and will be conducted at endline (2018) with a new cross-sectional sample of households and women within the households. The same households are not followed over time.

## Patient and public involvement

The development of the research question and outcome measures was informed by key stakeholders and patients' experience and preferences derived from free list responses, key informant interviews and focus group discussions conducted during the formative evaluation.[24–26] Input from key stakeholders and community members helped to ensure that the intervention would be responsive to community standards of acceptability and a feasible option to increase facility deliveries. Patients were not involved in study design, recruitment and/or conduct of the trial. Given the nature of the intervention, there was limited potential burden on patients, and therefore the burden of the randomised controlled trial was not assessed by the patients.

The primary audience for this evaluation is the Government of Zambia, particularly the Ministry of Health, Ministry of Community Development and the Ministry of Chiefs and Traditional Affairs, which will use the results to inform the development of maternal and child health strategies and policies in Zambia. We have disseminated the baseline findings to key stakeholders internal to Zambia and will disseminate the full study findings after endline. Many of the findings will likely be of broader interest throughout the region and globally where maternal mortality is high, resources are low and access to facility-based delivery remains an issue. As such, results of this evaluation will be disseminated as widely as possible through open-access journals, websites and international conferences.

## Procedures
### Data collection

At baseline and endline, a local team of enumerators literate in the appropriate local language(s) and in English are trained in qualitative and quantitative research methods and human subjects' protection. Surveys are designed in SurveyCTO Collect software (V.2.212; Dobility) and are captured electronically using encrypted tablets. The IDIs are digitally captured on audio recorders. Enumerators explain the tablet system to all participants and explain the digital audio recorders to those selected for IDIs.

Several checks assure the quality of collected survey data. First, enumerators participate in an extensive 5-day training. Second, the enumerators are overseen by data collection team leads with greater experience in data collection fieldwork. Team leads are overseen by a field supervisor. Team leads and the field supervisors review surveys for accuracy and completeness nightly. Third, field supervisors randomly select a 5% subsample of households to be audited; the auditor revisits these households and repeats a subset of survey questions that are checked for reliability. Fourth, the field supervisors conduct a short nightly debrief with the data leads who each oversee three other enumerators and are responsible for conducting the IDIs. Debriefs cover the following topics: field challenges, sampling, total surveys conducted and IDIs. Lastly, quantitative data are encrypted, uploaded and transferred nightly to the data analysis team where progress is reviewed in real time. On a nightly basis, qualitative data are removed from the recorders and saved on a password-protected computer.

### Data management

Survey data are captured on tablets and saved to the internal memory. During data collection, each evening, the field supervisor reviews the survey and encrypts it so data are no longer accessible on the tablet. The supervisor uploads encrypted data nightly to a secure server administered by SurveyCTO (V.2.212; Dobility). The evaluation team downloads the encrypted data using the SurveyCTO Client software (V.2.212; Dobility), and decrypts the data using a decryption key generated by the research team.

The evaluation team oversees data entry, management and storage for qualitative data. All IDIs are translated into English and transcribed verbatim. Digital recorders and paper copies of written notes are kept in a locked cabinet until transcriptions are checked for accuracy and completeness, at which point audio files are deleted and notes are shredded. The electronic transcriptions do not contain identifying information, only a study ID number linked to the quantitative survey. A separate linking file for the quantitative and qualitative data is password protected and only accessible to the study team.

### Data analysis

The primary independent variable of interest is assignment to the intervention. For the analysis, we will

compare baseline characteristics between the intervention and control groups to assess balance. We collect data on potential confounders to increase precision, analyse heterogeneity and, if necessary, control for any potential imbalance between the groups.

The primary dependent variable is the probability of facility delivery for most recent birth, based on self-report by mothers. Secondary outcomes include:

▶ Use of MWHs for antenatal care, delivery or postnatal services.
▶ Delivery by caesarean section.
▶ Maternal death.
▶ Neonatal death.

Because the data were self-reported and asked about experience up to 12 months before, there are limitations to what can reasonably be asked without introducing major recall bias. The survey captures additional indicators of morbidity including intravenous antibiotics, blood transfusions and referral to CEmONC, but we have limited secondary outcomes to those most likely to be clearly remembered.

All quantitative analyses will be conducted in SAS V.9.4 (SAS). Our quantitative analytic plan is three-fold, yielding descriptive, bivariate and multivariate statistics. First, we will describe the study sample, stratifying by intervention and control group and testing for differences between the groups. Second, we will estimate differences between the groups for primary and secondary outcomes, controlling for a set of baseline demographics. Categorical variables will be compared between the groups using a $X^2$ test when cell sizes are sufficient or Fisher's exact test when the cell sizes are small; continuous variables will be compared using t-tests if normally distributed or non-parametric Wilcoxon rank-sum tests if the distribution is non-normal. Third, we will fit several regression models to estimate the impact of the intervention on the primary and secondary outcomes, adjusting for baseline values, assignment matching variables and any imbalanced covariates. To control for the phased timing of implementation, we include a variable in the main models that captures the month the home opened.

All qualitative data will be analysed in NVivo V.10© software (QSR International). We will conduct a content analysis of the IDI transcripts. Coding themes have been identified a priori. Additional themes will be included as they emerge. We will triangulate findings with the quantitative data to identify consistencies, inconsistencies or additional themes to be explored. We will use the themes developed during the baseline analysis to analyse the endline data and identify any new themes as they emerge.

To systematically assess confounders and the risk of bias at the preintervention phase, intervention phase and postintervention phase, we will use the ROBINS-I tool.[36] This tool enables us to transparently report threats to validity of this quasiexperimental study during analysis, interpretation and dissemination. Results for

the primary and each secondary evaluation question will be presented.

## ETHICS
### Ethics approval and consent to participate
#### Ethical review boards
Prior to participant enrolment, ethical approvals were obtained from the Boston University Institutional Review Board (IRB), University of Michigan IRB (for a deidentified data set only) and the ERES Converge Research IRB, a private local ethics board in Zambia. We also obtained official approval to proceed with the study from the Zambia National Health Research Authority, which is responsible for oversight of all research conducted in the country. Adverse events, unanticipated problems and any protocol changes are reported to the IRBs and the Zambia National Health Research Authority per their guidelines, and all investigators are informed.

#### Potential risks and protections
This study poses minimal risk to study participants and several steps were taken to minimise risk and burden. To reduce the risk of disclosure of personal or sensitive information enumerators are trained to stop participants from disclosing information that is too sensitive. Participation may cause some discomfort from answering certain questions, particularly if the maternal or neonatal health outcomes were adverse. Enumerators are trained to minimise any potential discomfort or harm to all participants during all study activities to the greatest extent possible. We minimise any waiting by participants by scheduling meetings during times convenient to participants and interviews are kept to as short of time as possible taking breaks if necessary.

#### Potential benefits
There are no direct individual benefits to participating in the study. The evaluation results will generate evidence on the impact of MWHs on facility delivery for those who live farthest away. Findings will provide insight for policymakers into how, if found to be effective, MWHs can be part of a broader strategy to improve maternal and neonatal health outcomes.

#### Participant confidentiality
Throughout the study, we take care to ensure the confidentiality of data obtained from study participants. The HHS and IDIs are carried out in participants' private homes or somewhere the participant feels comfortable. We do not proceed with data collection until we can confirm that the location is acceptable and participants agree that they feel comfortable discussing study topics.

The linking file with identifiable data and basic demographics is stored in a separate file within the tablet system. On completion of data collection, all files are stored on a secure server during data analysis and dissemination. Only BU/RTC investigators have access to identifiable data. All analyses by study partners are conducted

on deidentified data sets per IRB approvals. Results are presented in aggregate format in technical reports to stakeholders and in manuscripts submitted for publication in scientific journals. Under no circumstances do organisations or individuals have access to the participants' individual demographic information or potential identifying information (job title, age range, sex and village). As explained above, the qualitative data are deidentified, linked to demographics only by a unique number.

### Informed consent

Prior to any data collection, we discuss the purpose of the study with local leaders so that the study activities are clearly understood. If a household is eligible, the study team proceeds with the informed voluntary consent process from the household head or the most senior woman in the household. The enumerators introduce themselves, the purpose of the study and explain what we are asking of them in terms of procedures, the risks and benefits, the right to withdraw without penalty at any time and confidentiality protections. Participants are informed that the alternative is to not participate in the study. The study team slowly and clearly asks for consent to participate. If a selected household participant declines participation, the next household on the randomly ordered list of eligible households is contacted. If a household head or senior woman consents to participating, the study team documents written informed consent and proceeds with the interview. The same process is used to consent the woman selected from within the household to respond to the survey; in some cases, this may be the same person as the household head or senior woman. A maximum of two individuals are consented per household.

We anticipate about 15% of the sample in each round to be between 15 and 17 years of age. In Zambia, 'emancipated minors' can enrol if they provide assent and their guardian or husband also provides consent. If a woman's husband is 18 or older, then he can provide informed consent on behalf of his wife; however, if he is also under 18 years old, then her legal guardian must provide consent. If under 18, the research team will allow the woman to first determine if she wishes to join the study (assent is provided) and then obtain consent by the guardian or husband. Thus, the individual's wishes are protected and she can determine if she wishes to be part of the study.

All informed consent or assent/consent is documented with a signature; in the event a participant cannot write, a witness signs the informed consent. A participant retains a copy of the informed consent form. The informed consent and assent processes are always conducted in the language most preferred by the participant.

### Costs and payments

For all activities, the participants volunteer only the time taken to complete this survey. There is no cash payment provided to participants for any portion of the study.

Participants receive pieces of fabric as small tokens of appreciation in recognition of their time and opportunity costs, in line with local IRB procedures.

### Limitations

This study has several limitations. First, half of study clusters could not be randomly assigned to either the intervention or control group due to political constraints and concern by the government about community fatigue. The selection bias resulting from the different assignment strategies is partially mitigated by ensuring comparison sites are matched on the same criteria as the other sites. Additionally, because one partner's comparison sites include existing MWHs as part of standard of care, and the other partner excluded sites with existing MWHs, we will analyse the full sample as a quasiexperimental CBA study and we will estimate the impact in both the non-randomised and randomised subsamples to assess potential bias introduced by non-random assignment and the differences in comparison site selection. Second, we limited household eligibility to those living at least 10 km from the health facilities and will not be able to assess impact on women living nearer to facilities. However, remote women are the primary target of the MWH model and stand to benefit the most from the intervention. To manage this limitation, in separate process evaluation protocols, each partner is collecting facility-based data to understand any changes in demographics among those utilising facilities for delivery. Lastly, because there are two implementing partners, there is a risk that the MWH model will be implemented differently across the sites. To mitigate this risk, we have developed and agreed on the precise elements of the MWH model based on both partners' formative research[24–26] and will be assessing implementation fidelity using harmonised tools in the companion process evaluation protocols.

**Author affiliations**
[1]Department of Global Health, Boston University School of Public Health, Boston, Massachusetts, USA
[2]Department of Pediatrics, Boston University School of Medicine, Boston, Massachusetts, USA
[3]Right to Care Zambia, Lusaka, Zambia
[4]National Health Research Authority, Pediatric Centre of Excellence, Lusaka, Zambia
[5]Department of Health Behavior and Biological Sciences, University of Michigan School of Nursing, Ann Arbor, Michigan, USA
[6]Center for Global Affairs and PAHO/WHO Collaborating Center, School of Nursing, University of Michigan, Ann Arbor, Michigan, USA
[7]Section of Infectious Diseases, Department of Medicine, Boston University, Boston, Massachusetts, USA

**Contributors** NAS led the scientific design, implementation, and was primarily responsible for drafting this manuscript. JK contributed to the development of the protocol, led the development of the study sample, coordinated data collection and contributed to revisions of the manuscript. TV and RB contributed to the revisions and science of the protocol and data collection instruments. RF contributed to the development of the protocol, implemented the electronic data capture system and contributed to the revisions of this manuscript. TN, GB, CJB and JRL provided feedback on the protocol and reviewed and edited the final manuscript. DHH provided scientific support, technical input into the survey design, and critically reviewed and edited the final manuscript. PCR helped conceptualise the scientific design of the study, provided technical input into the survey design, sampling

approach, and critically reviewed and edited the final manuscript. All authors read and approved the final version of the manuscript.

**Funding** This programme was developed and is being implemented in collaboration with MSD for Mothers, MSD's 10-year, $500 million initiative to help create a world where no woman dies giving life. MSD for Mothers is an initiative of Merck & Co, Kenilworth, NJ, USA (MRK 1846-06500.COL). The development of this article was additionally supported in part by the Bill & Melinda Gates Foundation (OPP1130329) (https://www.gatesfoundation.org/How-We-Work/Quick-Links/Grants-Database/Grants/2015/07/OPP1130329) and The ELMA Foundation (ELMA-15-F0017) (http://www.elmaphilanthropies.org/the-elma-foundation/).

**Disclaimer** The funders had no role in study design, data collection and analysis, decision to publish or preparation of the manuscript. The content is solely the responsibility of the authors and does not reflect positions or policies of MSD, the Bill & Melinda Gates Foundation or The ELMA Foundation.

**Competing interests** None declared.

**Patient consent** Obtained.

**Ethics approval** Boston University Institutional Review Board, University of Michigan Institutional Review Board, and ERES Converge Institutional Review Board in Zambia.

**Provenance and peer review** Not commissioned; externally peer reviewed.

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
