## [Reviewer comments · BMJ Open]

ARTICLE DETAILS

TITLE (PROVISIONAL)	Impact of maternity waiting homes on facility delivery among remote households in Zambia: protocol for a quasi-experimental, mixed-methods study
AUTHORS	Scott, Nancy; Kaiser, Jeanette; Vian, Taryn; Bonawitz, Rachael; Fong, Rachel; Ngoma, Thandiwe; Biemba, Godfrey; Boyd, Carol; Lori, Jody; Hamer, D.; Rockers, Peter

VERSION 1 – REVIEW

REVIEWER	Dr. Nafisa Lira Huq Assistant Scientist, icddr,b, Bangladesh
REVIEW RETURNED	28-Feb-2018

GENERAL COMMENTS	Row	Page	Comments
		24	5
	12	6	If BMJ has no word limitation the intervention needs to be more elaborated, otherwise it would be difficult to relate with the following sections of the protocol. In addition, if MWHs initiative is successful how it would be replicable to other low resource countries cannot be answered
	12-15	9	Controversial description between intervention and control groups. The construction of sentences needs to be clearer on which group is consisting of what.
	25	9	The selection process due to political constraints needs to be clarified to understand its rationale for non-randomization of 20 clusters
	19	10	Does the MWHs has the facility to store the drugs properly
	15	12	Generally the sample size for qualitative methods is flexible, however 10% sample for the qualitative component (240) seems to be huge for the impact study. Especially when the impact

			will be measured on quantitative analysis
	24	12	Again without a clear description about the MWHs it is difficult to understand what quality of care will be perceived by the beneficiaries
	1-2	15	Wondering how it was possible to conduct the 30 minutes IDI immediate after 45-30 minutes questionnaire interview. Had this effort faced any challenge, needs to be clarified
	34-35	15	Does the survey questionnaire pretested for finalization
	21-22	16	Quality check for the survey data is detailed out but this is absent for the qualitative data, especially there is no description of debriefing session which is mandatory for qualitative techniques
	27-28	17	The study design does not have the power to estimate and compare outcomes like maternal and neonatal deaths. The outcomes can look at other severe adverse maternal and neonatal outcomes rather than death. Regression models have been mentioned but the data analysis should describe how the maternal and neonatal outcomes will be compared between intervention and control groups
	17-30	18	Care should be taken for using the tense in a sentence while the enumerators were already trained and completed the baseline survey. There are some controversial description about the interview time and interval with that of the data collection section. Moreover providing cash of even small amount would bias the interview procedure which is unethical.
	30	20	Cost and payment section is contradicting with the above section of consent procedure, where 1-2 USD is mentioned

REVIEWER	Dr Sialubanje Cephas (MBCbB, MPH, PhD) Chainama College of Health Sciences, Lusaka, Zambia
REVIEW RETURNED	01-Mar-2018

GENERAL COMMENTS	This is a protocol for a study that aims to measure the impact of maternity waiting homes on facility delivery among remote households in Zambia. The protocol is generally well written, concise and easy to read. However, it needs some revision before it can be considered for publication. 1) Throughout the document, it is not clear whether the protocol is reporting a planned or ongoing study. The tense keeps changing
---

	from past, present and in some instances to future tense. This is confusing to the reader. There is need for clarity and consistency. 3) Dates for the study: Not clear; I am not sure the authors included them 4) Abstract: a) Analysis....it is not clear how the data will be analysed. b) Conclusion: Contrary to guidelines on reporting study protocols, the authors included the conclusion sections in both the abstract and main document. This should be removed. Reading through the conclusion in both sections, I noticed that the content ("To the best of our knowledge"....."This study will generate....") is actually a justification of the study. Let the authors remove the conclusion and take this content to the relevant section/under study justification. 5) In-text citations. Throughout the manuscript this needs attention. For example, the full stop should appear after the citation, and not before. Eg "...70 deaths per 100,000 live births by 2030.[1] Zambia's MMR is...." should be written as "...70 deaths per 100,000 live births by 2030 [1]. Zambia's MMR is...." 6) Page 6 line 41: Methods and Analysis. should read "Methods". I guess analysis is part of the methods and should appear as a subheading under the Methods section. better still, it should read data analysis. Page 9 line 15: signal function (i) should read birth attendant or staff and not "on staff". Signal function (v): it is not clear what the authors mean by travel time. Let the authors clarify on mode of travel (eg by car, bicycle, oxcart, etc) as the mode of travel determines the travel time 7) Introduction: Page 5 lines 24-32: There is a lot of repetition ..."MWHs is repeated several times.. 8) Sampling techniques: Page 12 line 54: Much as the authors make it clear that they used multi-stage sampling techniques, it is not clear how they randomly sampled the 10 villages from each catchment area. Did they have a pre-existing list of villages per catchment area from which they randomly sampled the 10 villages? Were the villages similar, geographically, etc? What assumptions did they make? 9) Typo and grammatical errors: There are a number of typo and grammatical errors in the document such as "comprised of" instead of "consisted of" or "comprised"(page 8 line 8); "antenatal instead of antenatal care"; fathest rather than farthest (page 6 line 31). 10) Page 14: Line 42: "Quality and completeness" should probably read as "accuracy and completeness" as these two are both part of quality! 11) Limitations: Page 20 line 6-7: "...half of study clusters could not be randomly assigned to either the intervention or control group due to political constraints". It is not clear what these political constraints are/were. Let the authors clarify this.
--	--

REVIEWER	Ariadna García Prado Public University of Navarra, Spain
REVIEW RETURNED	02-Mar-2018

GENERAL COMMENTS	This is a good study protocol. The research questions are interesting and can be very useful not only for policy-makers in Zambia but also for other countries that are exploring the possibility of using Maternity Waiting Homes (MWHs). Although other studies have highlighted the relevance of this strategy for promoting institutional birth and postpartum care, analyzing, in addition, aspects such as financial and managerial factors related to MWHs, this study protocol is intended to measure impact on health outcomes, which
---

	has not been done before. I have few comments that may help to improve the design and the study:  1. How are selected the 20 clusters that are randomly assigned to treatment and control group (10 to each)? Which is the total sample (how many clusters) from where you choose these 20 and how do you choose them? 2. Regarding the other 20 clusters that are assigned to treatment and control group without randomization: how were they selected in the first place? Was randomization used to select them? The paper says that these 20 clusters were assigned to control and treatment groups without randomization due to political constraints: it would be relevant to know what are the criteria followed to select those clusters that go to the treatment group in order to understand better what is the nature of the bias incurred. Is it based on poverty levels? Is based on number of inhabitants? It is important to make this transparent. 3. I understand that the sample is conformed by women who have delivered a baby in the last 12 months. However, it is not clear to me if these women have delivered in a health care facility, after using Maternity waiting homes or not. If the study is measuring the probability of using maternity waiting homes (and probability of facility delivery), it is difficult to know what is the intention to use them among women that have just delivered a baby if they have not used the Maternity Waiting homes. Women who have used maternity waiting homes and had an institutional birth would be an interesting sample to explore since they may decide, based on their experience, if they want to repeat or not. All these questions should be clarified. 4. I wonder if there is going to be an advertising strategy about the new Maternity waiting homes, so in case the women interviewed have not used them, at least, have heard of them and can say whether is their intention to use them or not. This would be useful not only for the research, but also in operational terms to increase the use of the Maternity Waiting Homes. 5. Finally, impact on health outcomes is going to be measured. In page 17 you talk about primary and secondary outcomes.  a. I wonder why you include as a secondary outcome delivery by c-section. Explaining the choice of secondary outcomes would be convenient. b. Maternal death and neonatal death can be included as outcomes (but not maternal mortality rate nor neonatal mortality rate because of the sample size and the short period of analysis: 18 months). However, I wonder if it is possible to include some morbidity indicators related to childbirth. Also, related to neonatal deaths I wonder if they are properly registered in Zambia. In some cultures newborn babies are not registered and their death is not registered. c. It would be interesting to measure the number of institutional births by women who used MWHs, versus the number of institutional births by women who did not use MWHs. 6. References could be completed with the following:  a. Penn-Kekana and others, 2017 (published at BMC pregnancy and childbirth) b. Fogliati et al, 2017 (published at Health policy and planning) c. Garcia-Prado and Cortez, 2012 (published at International Journal of Health Planning and Management)
--	--

REVIEWER	Tienke Vermeiden
-----------------	------------------

	Department of Health Sciences, Global Health, University Medical Centre Groningen/University of Groningen, The Netherlands Residing in Zimbabwe
REVIEW RETURNED	06-Mar-2018

GENERAL COMMENTS	This is well written study protocol on a topic of interest to those involved in maternal health in low- and middle income countries. The authors are correct that rigorous evidence on maternity waiting homes is needed and it is of great value that such a study is being implemented. I recommend to accept the protocol with minor revisions. I have added some comments to the attached PDF document. Some minor comments: 1) The authors speak of possible confounders, but do not provide much detail. They could consider reporting using tROBINS-I tool (Risk Of Bias In Non-randomized Studies - of Interventions). This will also allow them to provide arguments on why they call it a rigorous controlled before and after study. 2) The MWH model does not seem to include promotion of the intervention in the community, but their secondary evaluation questions include whether awareness and perceptions have changed over time in the MWH model sites. If the model does not include promotion/communication to the target group, how are women supposed to know about them? 3) It is not clear to me whether the MWH sites all had the model implemented at the same time. Otherwise, this will have an affect on the outcome measures. 4) The reason for having two sets of eligibility criteria for the study sites is unclear for me. 5) In the introduction, not all evidence on MWH effectiveness has been included.
--

VERSION 1 – AUTHOR RESPONSE

Response to reviewers: Manuscript ID bmjopen-2018-022224

Title: “Impact of maternity waiting homes on facility delivery among remote households in Zambia: protocol for a quasi-experimental, mixed-methods study”

Corresponding: Nancy Scott

Please find in the table below, a point by point response to the thoughtful comments from reviewers.

Responses to Comments from Reviewer # 1:

Comment #	Row/line	Page	Comments	Response

1	24	5	Is MWH a government policy of Zambia? There is no reference while describing it in	Thank you for this question. Maternity waiting homes (MHW) are one of Zambia's Strategy to reduce maternal mortality as indicated in the National Health Strategic Plan 2017-2021. Earlier, in 2013 the government mentioned MWHs in its Roadmap for Accelerating Reduction of Maternal, Newborn, and Child Mortality. The Zambian government recognizes MWHs as a method for achieving greater access to skilled birth attendance. To address this comment, we have included wording within the description of the intervention (p. 9) stating that although there is no specific policy on MWHs, the government of Zambia supports the construction and use of MWHs and has taken MWHs as one of its strategies to increase access to skilled deliveries; contributing to its goal of reducing maternal mortality as articulated in the National Health Strategic Plan 2017-2021.
2	12	6	If BMJ has no word limitation the intervention needs to be more	Thank you for this comment. We have elaborated on the MWH intervention being evaluated, inclusive of additional

			elaborated, otherwise it would be difficult to relate with the following sections of the protocol. In addition, if MWHs initiative is successful how it would be replicable to other low resource countries cannot be answered	references explaining how they were designed. (p. 6-7) In response to the reviewer's comment on replicability, we are conducting a parallel process evaluation which will generate evidence to scale the intervention, if demonstrated to be effective in this impact evaluation.
3	12-15	9	Controversial description between intervention and control groups. The construction of sentences needs to be clearer on which group is consisting of what.	We have better differentiated between intervention and control, clarifying what each group consists of. (p. 9)
4	25	9	The selection process due to political constraints needs to be clarified to understand its rationale for non-randomization of 20 clusters	Thank you for this comment. We have made revisions to clarify this within the document. When the partner organization approached the Ministry of Health about conducting this study, the Ministry was very reluctant to allow the partner to randomly select intervention sites as other organizations were conducting projects and research

			within the districts. The Ministry feared community fatigue if the project began constructing or collecting data at health facilities where other large projects existed. Therefore, the Ministry and the partner organization
--	--	--	--

				worked collaboratively to identify intervention sites where community fatigue was unlikely to occur and to then match them with comparison sites. (p. 11)
5	19	10	Does the MWHs has the facility to store the drugs properly?	The MWHs are not intended for clinical care, as clarified in the intervention section (p. 7) and therefore do not store drugs. All drugs are stored properly at the adjacent health facility.
6	15	12	Generally the sample size for qualitative methods is flexible, however 10% sample for the qualitative component (240) seems to be huge for the impact study. Especially when the impact will be measured on quantitative	We agree that it is a large qualitative sample. We selected a 10% sub-sample to better explore the context. Because use the seven districts are spread out, we wanted to ensure we reached saturation in each district. We have more clearly justified the rationale for the qualitative sample size in the revisions. (p. 13)

			analysis	
7	24	12	Again without a clear description about the MWHs it is difficult to understand what quality of care will be perceived by the beneficiaries	We hope that changes we made in response to comment 2 above clarify the quality constructs about which beneficiaries will provide their perceptions. The MWH model was designed to be responsive to community- defined expectations, standard of acceptability and their perceptions of quality including safety, comfort, management and services. (p. 6)
8	1-2	15	Wondering how it was possible to conduct the 30 minutes IDI immediate after 45- 30 minutes questionnaire interview. Had this effort faced any challenge, needs to be clarified	We appreciate this comment. In the baseline, we did not experience any challenges with this, barring the small proportion who opted out of the IDI after being selected. The consent forms were clear on the time necessary for completing It is not uncommon for a household survey in Zambia to last between 1 – 1 ½ hours. We did not change the manuscript in response to this comment.
9	34-35	15	Does the survey questionnaire pretested for finalization	Yes, the survey questionnaire is pre-tested among 50 respondents at each baseline and endline. At baseline, as

				expected, small adjustments were made in response to the pre-test, mostly changing more formal translations into the vernacular. There were no major changes required. We have clarified this in the manuscript. (p. 11)
10	21-22	16	Quality check for the survey data is detailed out but this is absent for the qualitative data, especially there is no description of debriefing session which is mandatory for qualitative techniques	Thank you for this comment. We have elaborated on our qualitative data quality checks in the document (p. 17). We conducted short debriefing sessions with the data leads nightly, each of whom oversaw three data collectors and conducted IDIs. These debriefs covered the following topics: field challenges, sampling, total surveys conducted, and IDIs.
11	27-28	17	The study design does not have the power to estimate and compare outcomes like maternal and neonatal deaths. The outcomes can look at other severe adverse maternal and neonatal outcomes rather than	Thank you for this comment. It is similar to that of Reviewer 3, comment 5b. We agree that we do not have sufficient sample size to detect impacts on maternal and neonatal deaths (primary outcome is facility delivery), though we will measure and report on the prevalence of these outcomes.

		death. Regression models have been mentioned but the data	For maternal outcomes, we explore impacts on self-reported overall health and proxies for complications that
--	--	---	--

		analysis should describe how the maternal and neonatal outcomes will be compared between intervention and control groups	we determined may be reasonably remembered, including Caesarean section, IV antibiotics, blood transfusions, and referral to CEmONC. We have clarified this on p. 19 For neonatal outcomes, we explore impacts on vaccination status, recent illness, and feeding methods (table 2, p 13). Differences in the mean values of these outcomes will be estimated, controlling for a set of baseline demographic variables (p. 19).
--	--	--	---

12	17-30	18	Care should be taken for using the tense in a sentence while the enumerators were already trained and completed the baseline survey. There are some controversial description about the interview time and interval with that of the data collection section. Moreover providing cash	Thank you for this comment. We have addressed the tense issue throughout the manuscript. Additionally, we have harmonized the conflicting wording on the interview time. Cash was not given to respondents. A token of appreciation, valued at \$1-2 (i.e: piece of fabric), was given in recognition of the respondent's time contributed to the interview, as is customary and required by the local IRB, in Zambia. This was approved by all three of the reviewing
----	-------	----	---	--

			of even small amount would bias the interview procedure which is unethical.	ethical boards. We have clarified that pieces of fabric were given to respondents in recognition of their time (p. 23).
13	30	20	Cost and payment section is contradicting with the above section of consent procedure, where 1-2 USD is mentioned	Thank you for this comment. We have clarified that it is a token in recognition of their time. We have moved this to the cost and payments section of the paper (p. 23).
Responses to Comments from Reviewer # 2:				
1			Throughout the document, it is not clear whether the protocol is reporting a planned or ongoing study. The tense keeps changing from past, present and in some instances to future tense. This is confusing to the reader. There is need for clarity and consistency.	Thank you for this comment. We agree, as it was difficult to write this when baseline had already occurred and endline is in the future. Per our response to the comment 12 from Reviewer #1 above, we have reread the manuscript and adjusted tense where appropriate. We hope this provides clarity for the reader.
2.			Dates for the study: Not clear; I am not sure the authors included them	Thank you for noticing this. The dates were originally included on P 9, line 49-54. We have also added it to the study setting section on p 8.

4a	Abstract	Abstract: Analysis....it is not clear how the data will be analysed.	We have clarified in the abstract that we will calculate descriptive statistics and adjusted odds ratios; qualitative data will be analyzed using content analysis.
4b	Abstract	b) Conclusion: Contrary to guidelines on reporting study protocols, the authors included the conclusion sections in both the abstract and main document. This should be removed. Reading through the conclusion in both sections, I noticed that the content ("To the best of our knowledge"..... "This study will generate....") is actually a justification of the study. Let the authors remove the conclusion	We appreciate this comment. We have adjusted the protocol accordingly and removed the 'conclusions' section from both the abstract and from the main manuscript.

		and take this content to the relevant section/under study justification.	
5		In-text citations. Throughout the	Thank you for your suggestion. However, per submission

			manuscript this needs attention. For example, the full stop should appear after the citation, and not before. Eg "...70 deaths per 100,000 live births by 2030.[1] Zambia's MMR is...." should be written as "...70 deaths per 100,000 live births by 2030 [1]. Zambia's MMR is...."	guidelines from BMJ Open, "Reference numbers in the text should be inserted immediately after punctuation (with no word spacing)." As such, we have not adjusted the document in response to this comment.
6.	41	6	Page 6 line 41: Methods and Analysis. should read "Methods". I guess analysis is part of the methods and should appear as a subheading under the Methods section. better still,it should read data analysis.	We have changed the heading to read "Methods" (p. 8) and better clarified the sub-headings.
6.	15	9	signal function (i) should read birth attendant or staff and not "on staff". Signal function (v): it is not clear what the authors mean by travel time. Let the	We have not adjusted signal function (i) per your suggestion because our intention is to ensure that there is a birth attendant on the staff and employed at the health center.

		authors clarify on mode of travel (eg by car, bicycle, oxcart, etc) as the mode of travel determines the travel time	Travel time to CEmONC is an indicator reported by the government of Zambia in its health facility assessment, the only data available at the time of selection. We have clarified that in this report, they use driving time on p. 10.
7.		Introduction: Page 5 lines 24-32: There is a lot of repetition ... "MWHs is repeated several times..	We have adapted the paragraph to make it less repetitive (p. 5).
8.		Sampling techniques: Page 12 line 54: Much as the authors make it clear that they used multi-stage sampling techniques, it is not clear how they randomly sampled the 10 villages from each catchment area. Did they have a pre-existing list of villages per catchment area from which they randomly sampled the 10 villages? Were the villages	We appreciate this question. We have expanded upon and clarified our sampling methods within the document on p. 14-15. The explanation is also summarized below: We first develop a list of all villages within each catchment area through consultation with health facility staff. We then visit each village and record GPS coordinates for each. Next, we calculated the travel distance for each village and include only those village more than 10km from the health facility in our sample frame. We then randomly select approximately 10 villages from each catchment area. We randomly select them using probability proportionate to

		similar, geographically, etc? What assumptions did they make?	size by listing the population count of each village (i.e.: if village 1 had 30 people, 1-30; village 2 had 20 inhabitants, 31-50), then use the random number generator function in Excel to select the villages. If the village selected do not have sufficient numbers of eligible women (n=6), we then select the next village in the draw. For each selected village, we list all households and visit each to determine eligibility for the study. We randomly order this list of households and visit each household in that order. If more than one woman is eligible in the
--	--	---	--

			household, the electronic data capture system is programmed to randomly select a respondent. This process assumes that the health facility staff are able to accurately and completely identify all villages within their catchment area.
--	--	--	---

9.		Typo and grammatical errors: There are a number of typo and grammatical errors in the document such as "comprised of" instead of "consisted of" or	We have addressed the three errors noted in the comment and have checked for any additional grammatical errors.
----	--	--	--

			"comprised"(page 8 line 8); "antenatal instead of antenatal care";fathest rather than farthest (page 6 line 31).	
10.	42	14	Page 14: Line 42: "Quality and completeness" should probably read as "accuracy and completeness" as these two are both part of quality!	We have made your suggested edit. (p. 17)
11.	6-7	20	Limitations: Page 20 line 6-7: "...half of study clusters could not be randomly assigned to either the intervention or control group due to political constraints". It is not clear what these political constraints are/were. Let the authors clarify this.	Thank you for this comment. Per our response to comment 4 by Reviewer 1 and this comment, we have made revisions to clarify this randomization in the methods section (p. 10) and the limitations section (p. 23-24). Specifically, when the partner organization approached the Ministry of Health about conducting this study, the Ministry was reluctant to allow the partner to randomly select sites within the chosen districts as other organizations were also conducting projects and research. The Ministry feared community fatigue if the project began constructing or collecting data at health facilities where other large projects existed. Therefore, the Ministry and

				the partner organization worked collaboratively to identify sites where this community fatigue was unlikely to occur and match them to comparison sites.
Responses to Comments from Reviewer # 3:				
1.			How are selected the 20 clusters that are randomly assigned to treatment and control group (10 to each)? Which is the total sample (how many clusters) from where you choose these 20 and how do you choose them?	Thank you for requesting clarity on this section. We have addressed your comment by expanding on the selection process on p. 10 of the manuscript. To select the 40 sites (20 per partner), one partner selected the 20 farthest away, then matched on volume and distance, then randomly assigned matched pairs to intervention or control, using the RAND function in Excel. The other partner, worked with the government to identify 10 intervention sites. They then selected an additional 10 facilities as comparisons, matched on distance from CEmONC and delivery volume.
2.			Regarding the other 20 clusters that are assigned to treatment	We have made revisions to clarify this within the document. Please see the above responses to comment 4

			and control group without randomization: how were they selected in the first place? Was randomization used to select	from Reviewer 1, and comment 11 from Reviewer 2, to better understand the randomization process and the limitations faced by the partner that was unable to
--	--	--	---	--

			them? The paper says that these 20 clusters were assigned to control and treatment groups without randomization due to political constraints: it would be relevant to know what are the criteria followed to select those clusters that go to the treatment group in order to understand better what is the nature of the bias incurred. Is it based on poverty levels? Is based on .number of inhabitants? It is important to make this transparent.	randomiz e. We have made changes in the section (p. 10) and the limitations section. (p. 23-24)
--	--	--	--	--

3.			I understand that the sample is conformed by women who have	Thank you for this comment. We have addressed this comment in three ways. First, our aim is to estimate the
----	--	--	--	--

		delivered a baby in the last 12 months. However, it is not clear to me if these women have delivered in a health care facility, after using Maternity waiting homes or not. If the study is measuring the probability of using maternity waiting homes (and probability of facility delivery), it is difficult to know what is the intention to use them among women that have just delivered a baby if they have not used the Maternity Waiting homes. Women who have used maternity waiting homes and had an institutional birth would be an interesting sample to explore since they may decide, based on their experience, if	impact of the MWH intervention based on an intention-to-treat analysis, and for this we need to sample all women, irrespective of delivery location or whether they used a MWH. With this strategy, we will still be able to explore the relationship between use of the MWH and location of delivery. We have clarified this on p.14 of the manuscript. Second, the household survey captures intended delivery location and intention to utilize a MWH, so we will be able to explore this in the analysis. We have clarified this in table 2 on p. 12. Third, we agree with your suggestion that fruitful research questions may be studied with the sample of women who utilized the MWH and considering intention to repeat based on experience. We capture this, clarified in table 2. We will also explore this in more depth in the process evaluation mentioned on p. 24.
--	--	--	--

		they want to repeat or not. All these questions should be clarified.	
4.		I wonder if there is going to be an advertising strategy about the new Maternity waiting homes, so in case the women interviewed have not used them, at least, have heard of them and can say whether is their intention to use them or not. This would be useful not only for the research, but also in operational terms to increase the use of the Maternity Waiting Homes.	Thank you for this thoughtful comment. We agree that it is useful for operations as well as the research. Reviewer 4 raised the same issue in comment 2 below. clarified that the MWH model includes promotion of the intervention in the community through several mechanisms: health facilities at ANC, community health workers (SMAGs) and traditional leadership. clarified this on p. 7, when we expanded on the description of the intervention in response to other comments. Additionally, the household survey captures if a woman has heard of MWHs, from where she has heard of them, her previous utilization of them and her future intentions

				to utilize them. We have clarified this in table 2, p. 12.
5a.			Finally, impact on health outcomes is going to be	Thank you, we appreciate your suggestions in this and the following comment. We initially considered other

			measured. In page 17 you talk about primary and secondary outcomes. I wonder why you include as a secondary outcome delivery by c-section. Explaining the choice of secondary outcomes would be convenient.	morbidity outcomes, but because the data were self-reported and asked about experience up to 12 months before, there were limitations to what we thought we could reasonably ask without introducing major recall bias. While the survey captures other proxies for complications we can examine (IV antibiotics, blood transfusion and referral to CEmONC), we felt that delivery by caesarean section would be the most useful secondary outcome as it could be influenced by the MWH and has low susceptibility to recall bias. We have clarified this on p. 19.
--	--	--	---	---

5b.			b. Maternal death and neonatal death can be included as outcomes (but not maternal mortality rate nor neonatal mortality rate because of the	Thank you, we appreciate your suggestions. Please see comment 5a above. We will consider additional morbidity outcomes for future studies of the impact of MWHs. Additionally, we will be better able to assess morbidity indicators under our separate process
-----	--	--	---	---

		sample size and the short period of analysis: 18 months). However, I wonder if it is possible to include some morbidity indicators related to childbirth. Also, related to neonatal deaths I wonder if they are properly registered in Zambia. In some cultures newborn babies are not registered and their death is not registered.	evaluation protocols (p. 24). Regarding the second point, neonatal deaths are registered at health facilities and sometimes by village headmen but not all are captured. Our data collection system through a household survey should allow us to identify all maternal and neonatal mortality events among sampled participants. When determining the eligibility for a household, we ask about any deliveries within the previous 12 months, regardless of the current vital status of the mother or child (p. 15).
5c.		It would be interesting to measure the number of institutional births by women who used MWHs, versus the number of institutional births by women who did not use MWHs.	Thank you for this suggestion. We agree. Although we can compare those who used the MWH to those who did not from our sample, this is not a feasible way to estimate total institutional deliveries, and this protocol is not written to collect facility-level data. However, as mentioned on page 24, we have process evaluation protocols examining utilization of the MWHs which captures MWH and facility-

			based data and will be a better source of data for analysis.
6		a. Penn-Kekana and others, 2017 (published at BMC pregnancy and childbirth) b. Fogliati et al, 2017 (published at Health policy and planning) c. Garcia-Prado and Cortez, 2012 (published at International Journal of Health Planning and Management)	Thank you for your suggestion of additional literature for the introduction. We have included Fogliati et al, 2017 and Garcia-Prado and Cortez, 2012, as well as other relevant literature that has recently become available. Please note, we did not include Penn-Kekana et al, 2017 as this article discusses the facilitators and barriers to MWH implementation, not MWH effectiveness. It is more applicable to the process evaluation protocol.

1		The authors speak of possible confounders, but do not provide much detail. They could consider reporting using tROBINS-I tool (Risk Of Bias In Non-randomized Studies - of Interventions). This will also allow them to provide arguments on why they call it a rigorous controlled before and	Thank you for this suggestion as we were not aware of this tool. We will use the suggested tool to assess risk of bias as we report. We have adjusted the manuscript accordingly in the analysis section (p. 20).
---	--	---	--

			after study.	
2.			The MWH model does not seem to include promotion of the intervention in the community, but their secondary evaluation questions include whether awareness and perceptions have changed over time in the MWH model sites. If the model does not include promotion/communication to the target group, how are women supposed to know about them?	Thank you for this observation. The MWH model does include a promotion of the intervention in the community through several mechanisms. First, health facility staff promote the MWH at all ANC visits. Over 95% of women attend at least the first ANC visit, so most women are exposed at the health facility. Second, the Safe Motherhood Action Group members promote the use of MWHs during their routine outreach activities. Lastly, the traditional leadership (chiefs and headmen) actively promote the use of MWHs at their community meetings. We have clarified this on p. 7, when we expanded on the description of the intervention in response to other comments.
3.			It is not clear to me whether the MWH sites all had the model implemented at the same time. Otherwise, this will have an affect on the outcome	We appreciate this point. The reviewer is correct that there was some phasing of implementation due to the logistics of the construction process. We have mentioned this on p. 8. We will control for the timing of implementation by including a variable in our main models that captures the

		measures.	month the home opened. We have clarified this in the analysis section of the protocol on p. 19.
4.		The reason for having two sets of eligibility criteria for the study sites is unclear for me.	Thank you. We have clarified that while it is essential for the health facility to be able to manage complications, the data available across districts were inconsistent. Therefore, we established two sets of eligibility criteria; clusters were eligible if they: 1) were located within 2 hours transfer time to a referral hospital, 2) performed at minimum of 150 deliveries per year, and 3) met at least one of the two sets of conditions. This is now explained more clearly under a section entitled: Eligibility criteria of study clusters (p. 10).
5.		In the introduction, not all evidence on MWH effectiveness has been included.	Thank you for your comment. Additional articles have been included in the introduction, in response to reviewer #3's comments above.

VERSION 2 – REVIEW

REVIEWER	Ariadna García Prado Public University of Navarra, Department of Economics, Pamplona (Navarra), Spain
REVIEW RETURNED	26-Jun-2018

GENERAL COMMENTS	The authors have responded to my comments, clarifying my questions and improving the text accordingly. My only minor comment (not subject to acceptance) is related to page 11 (Selection and assignment of study clusters to study arm): Although the selection and assignment process is explained much better now, in the last part of that section is not clear to me yet how the Ministry of Health identified 10 intervention sites, i.e. which were the employed criteria to select those 10. The last sentence: " From the remaining eligible, they excluded those with an existing functional MWH (...)". Meaning that the Ministry of Health chose places where there were not MWH at all? or not functional MWH? Linked to this, I wonder if the evaluated MWH were built as part of the infrastructure component of the project, or were already in place but not ready to use yet. Clarifying the infrastructure component of the project will help to understand this last part of the Selection and assignment section.
--

VERSION 2 – AUTHOR RESPONSE

Thank you for the encouraging news. We have addressed the remaining reviewer comment by clarifying the existing infrastructure component and selection process in the 'Selection and Assignment of Study Clusters to Study Arm' section. We have also made minor edits to spelling and punctuation throughout the document. The marked-up and clean copies are both attached to this re-submission.